# Adenosine Triphosphate Accumulated Following Cerebral Ischemia Induces Neutrophil Extracellular Trap Formation

**DOI:** 10.3390/ijms21207668

**Published:** 2020-10-16

**Authors:** Seung-Woo Kim, Dashdulam Davaanyam, Song-I Seol, Hye-Kyung Lee, Hahnbie Lee, Ja-Kyeong Lee

**Affiliations:** 1Department of Anatomy, Inha University School of Medicine, Incheon 22212, Korea; mymaysun@hanmail.net (S.-W.K.); dashka010925@gmail.com (D.D.); ssie8878@naver.com (S.-I.S.); marrina@hanmail.net (H.-K.L.); 317360@inha.ac.kr (H.L.); 2Medical Research Center, Inha University School of Medicine, Incheon 22212, Korea; 3Department of Biomedical Sciences, Inha University School of Medicine, Incheon 22212, Korea; 4Program in Biomedical Science & Engineering, Inha University School of Medicine, Incheon 22212, Korea

**Keywords:** NETosis, stroke, ATP, PAD4, ROS

## Abstract

In ischemic stroke, neutrophils infiltrate damaged brain tissue immediately following the ischemic insult and aggravate inflammation via various mechanisms which include neutrophil extracellular traps (NETs) formation. In the present study, we showed that adenosine triphosphate (ATP), a DAMP molecule, accumulates in the brain and induces NETosis in brain parenchyma and in circulating neutrophils (PMNs) isolated from a murine model of stroke induced by middle cerebral artery occlusion (MCAO). Expression of peptidylarginine deiminase-4 (PAD4), which induces citrullination of histones H3 (CitH3) and initiates NETosis, was significantly enhanced in brain parenchyma and blood PMNs following MCAO. ATP or BzATP (a prototypic P2X7R agonist) significantly enhanced the inductions of PAD4 and CitH3 in a P2X7R-dependent manner and intracellular Ca^2+^ influx, PKCα activation, and NADPH oxidase-dependent reactive oxygen species (ROS) production play critical roles in this ATP-P2X7R-mediated NETosis. In our MCAO animal model, NETosis was markedly suppressed by treatment with apyrase, an enzyme hydrolyzing ATP, but enhanced by co-treatment of BzATP, confirming ATP-P2X7R-mediated NETosis. Since ATP not only induced NETosis but was also extruded after NETosis, our results indicate that ATP accumulated in the ischemic brain induces NETosis, mediating a cross-talk linking NETosis with neuronal damage that might aggravate inflammation and brain damage.

## 1. Introduction

Neutrophil extracellular trap (NET) formation, a process known as NETosis, is a unique type of neutrophil cell death that captures pathogens through the ejection of DNA and anti-microbial enzymes [1]. NETosis is not only induced by pathogens (e.g., bacteria, fungi, virus, and protozoa) but also by host factors such as activated platelets and proinflammatory cytokines [1,2,3]. As part of the inflammatory response to infection, histones are post-translationally modified in a process that includes citrullination by peptidylarginine deiminase 4 (PAD4), which induces chromatin decondensation [4]. Neutrophil elastase and myeloperoxidase (MPO) also promote chromatin decondensation and form extracellular traps composed of chromatin and proteases [2]. Under the conditions of sterile inflammation, activated platelets were shown to contribute to NET formation in transfusion-related acute lung injury [5]. Interleukin-33 released from liver sinusoidal endothelial cells aggravates liver damage during hepatic ischemia/reperfusion by inducing excessive inflammation and NETosis [6]. Recently, we reported that high mobility group box 1 (HMGB1), a danger associated molecular patterns (DAMPs), triggers NET formation and exacerbates neuronal damage in the post-ischemic brain [7].

NETosis has been reported as an important process involved in the infectious and non-infectious diseases of the central nervous system (CNS). NET-like structures and amplified neurotoxicity have been reported in the brain after IL-1β-induced transendothelial migration of neutrophils [8]. In a mouse model of Alzheimer’s disease, cells positive for citrullinated histone H3 (CitH3; a marker of NETosis) were detected in areas containing amyloid-β (Aβ) deposits [9]. Similarly, CitH3-positive neutrophils were found in perivascular spaces, in brain parenchyma near the blood vessels, and in lumen of the capillary in an animal model of permanent middle cerebral artery occlusion (MCAO) [10]. In our previous study, we have demonstrated an induction of NETosis by HMGB1 (a prototypical DAMP) in the post-ischemic brain and shown that suppression of NET formation mitigates delayed inflammation and vessel damage [7]. Recently, Kang et al. [11] also reported that NETs impair revascularization and vascular remodeling after stroke. Therefore, a growing body of evidence implicate NETosis in both infectious and non-infectious CNS diseases.

ATP released by necrotic cells was reported to act as a proinflammatory DAMP [12]. P2X7 receptor (P2X7R) is an ATP-gated cation channel that participates in various immune responses, including activation of the inflammasome, induction of inflammatory mediator, and production of free radicals [13,14]. Under pathophysiological conditions, ATP acts through P2X7R to induce ion flux, increasing the influx of Ca^2+^ and Na^+^ and efflux of K^+^, which enhances reactive oxygen species (ROS) production [15]. ATP also stimulates the secretion of proinflammatory cytokines, for example, inducing IL-1β secretion from neutrophils mediated by P2X7R-NLRP3 inflammasome [16] or from lipopolysaccharide (LPS)-treated human macrophages [13] and IL-6 and IL-18 secretion from monocytes via P2X7R [17,18]. ATP was recently reported to aggravate ischemic acute kidney injury, with increasing PAD4 expression in a mouse model of renal ischemia/reperfusion injury [19]. In addition, ATP was shown to amplify calcium ionophore- or PMA-induced NET formation but fail to induce NETosis for itself [20].

In the ischemic brain, plasma membrane disruption caused by necrotic death of neurons results in ATP leakage into the extracellular space [21]. Furthermore, ATP is released from the vesicles of neurons depolarized owing to an impairment in Na^+^-K^+^ pump function and from microglia and astrocytes by exocytosis [22] or through their cell membrane channels [23]. While extracellular ATP was reported to elicit a number of beneficial effects in the peripheral and central nervous systems through its neurotransmitter functions and its chemoattractant properties [24,25], it elicits detrimental effects through its binding to P2XR. In the present study, we investigated whether extracellular ATP released after cerebral ischemia induces NETosis and what are the underlying molecular mechanisms and signaling pathways. We explored the possibility that DAMPs are important factors inducing NETosis in non-infectious diseases.

## 2. Results

### 2.1. Induction of PAD4 and CitH3 in Plasma and Brain Parenchyma after MCAO and by ATP and BzATP in Normal Animal

Examination of ATP level after MCAO using an ATP luminescence kit showed a significant surge in plasma 12 h following MCAO (Figure 1A). Expression of PAD4, which induces citrullination of histones H3 (CitH3) and initiates NETosis, was found to be increased in PMNs isolated from peripheral blood (blood PMNs) following MCAO, peaking at 12 h (Figure 1B). Additionally, level of CitH3 (a NETosis marker) was also significantly increased at 12 h after MCAO in blood PMNs (Figure 1B), which is consistent with our previous report [7]. In brain parenchyma, robust inductions of PAD4 and CitH3 were also detected in cortical penumbra, where NETosis was shown to occur following MCAO [7], at 24 h after MCAO (Figure 1C), which is later than those in blood PMNs. It is notable that PAD4 level was slightly decreased until 12 h post-MCAO and then markedly increased at 24 h (Figure 1C). Importantly, when we administered BzATP (5 mg/kg, i.v.), a prototypic P2X7R agonist, at 6 h post-MCAO, levels of PAD4 and CitH3 in cortical penumbra were significantly enhanced in BzATP dose-dependent manner (Figure 1D), demonstrating P2X7R-mediated induction of NETosis in the ischemic brain. Moreover, administration of ATP (5, 10, 20, or 50 mg/kg, i.v.) or BzATP (5 mg/kg, i.v.) to normal animals increased PAD4 and CitH3 levels in circulating blood PMNs (Figure 1E,F). Taken together these results raise a possibility that extracellular ATP accumulated during the acute phase of ischemic stroke may play a role in the NETosis induction.

### 2.2. ATP Induced PAD4 and CitH3 Upregulation in Neutrophils and NETosis in a P2X7R-Dependent Manner

To determine whether ATP upregulates the expression of CitH3 and PAD4 and induces NET formation, we treated blood PMNs with ATP or BzATP. Blood PMNs treated with ATP (50 μM) or BzATP (50 μM) exhibited significantly increased CitH3 levels after 2 h of treatment (Figure 2A). Following treatment of blood PMNs with a range of concentrations of ATP or BzATP for 4 h, CitH3 induction was detected at 50 μM of ATP or 25 μM of BzATP (Figure 2B). Additionally, PAD4 levels were significantly increased following treatment of blood PMNs with ATP (100 μM) or BzATP (50 μM) for 2 h (Figure 2C,D), indicating that ATP or BzATP upregulated PAD4 and subsequently induced CitH3 in neutrophils. Significant inductions of CitH3 and PAD4 were also obtained in experiments evaluating PMNs derived from bone marrow (BM-PMNs) at different effective doses and times (Appendix A). These results prompted us to evaluate the importance of P2X7R, a purinergic receptor known to respond to ATP or BzATP [26]. Pretreatment of blood PMNs with A438079 (a P2X7R antagonist) at 10 or 20 μM for 20 min suppressed the induction of PAD4 by ATP or BzATP almost completely (Figure 2E,F). Moreover, induction of CitH3 by ATP or BzATP was also significantly suppressed by the pretreatment of blood PMNs with A438079, which proved to be as effective as Cl-amidine, a PAD inhibitor, especially in BzATP-treated cells (Figure 2G,H). Staining with Sytox Green, a high affinity nucleic acid dye, indicated that treatment with ATP or BzATP induces the extracellular release of dsDNA from blood PMNs (Figure 2I). When the amount of dsDNA in the culture medium (another NETosis marker) was measured, it was significantly increased in ATP dose-dependent manner in blood PMNs and as expected, pretreatment of blood PMNs with A438079 significantly suppressed it (Figure 2J). Amounts of extracellular DNA-neutrophil elastase (NE-DNA) complex were also significantly increased by ATP and it was suppressed by A438079 or Cl-amidine (Figure 2K), further confirming the results. Taken together, these results demonstrate that ATP and BzATP upregulate PAD4, as well as subsequent CitH3 induction and NET formation via P2X7R.

### 2.3. Intracellular Ca^2+^ Influx and PKCα Activation Are Involved in ATP or BzATP-Mediated NETosis

P2X7R is a Ca^2+^-permeable cationic channel [13] whose stimulation induces PKC activation and increases intracellular ROS production [27]. When we examined Ca^2+^ influx into neutrophils using Fura-2 AM (an intracellular indicator of Ca^2+^ levels), intracellular Ca^2+^ levels were significantly increased in BzATP-treated blood PMNs, with the increase being markedly reduced by pretreating the cells with A438079 (10 μM) or a cell-permeable Ca^2+^ chelator BAPTA-AM (10 μM) (Figure 3A). Pretreatment with BAPTA-AM suppressed ATP- and BzATP-induced upregulations of PAD4 and CitH3 (Figure 3B,C) as well as dsDNA release in culture media (Figure 3I), indicating that Ca^2+^ influx is crucial for ATP- or BzATP-induced NETosis. In addition, treatment of blood PMNs with 100 μM of ATP (elevated the levels of phosphorylated PKCα within 30 min (or 60 min with 50 μM of ATP) (Figure 3D), and these elevations were suppressed by pretreatment with A438079 or BAPTA for 20 min (Figure 3E,F), indicating that P2X7R and Ca^2+^ influx play critical role in PKCα activation. Interestingly, while pretreating blood PMNs with Gö6983 (a pan-PKC inhibitor) did not suppress Ca^2+^ influx (Figure 3A), BAPTA-AM pretreatment suppressed ATP-induced activations of pPKCα (Figure 3F), indicating that Ca^2+^ influx takes place upstream of PKCα activation. Importantly, pretreatment with Gö6983 at 1 or 2 μM for 20 min suppressed ATP- or BzATP-induced upregulations of PAD4 and CitH3 in blood PMNs (Figure 3G,H) as well as the amounts of extracellular DNA complexed with neutrophil elastas (NE-DNA) (Figure 3I). Suppression of dsDNA release by BAPTA or Gö6983 was further supported by Sytox Green staining (Figure 3J). Critical roles of Ca^2+^ influx and PKCα activation were confirmed in Bz-ATP-treated blood PMNs (Appendix A). Taken together, these results show that Ca^2+^ influx and PKCα activation play a crucial role in ATP- or BzATP-induced NETosis.

### 2.4. ROS Production Is Involved in ATP-Mediated Induction of NETosis

When we examined ROS production using CM-H2DCFDA (a general ROS indicator), the intensity of CM-H2DCFDA fluorescence was found to increase significantly in blood PMNs after ATP treatment, but was markedly suppressed by pretreating the cells with A438079 (10 μM), BAPTA-AM (10 μM), or Gö6983 (10 μM) (Figure 4A,B), suggesting that ATP induces ROS production in blood PMNs in a P2X7R-, Ca^2+^-, and PKC-dependent manner. Additionally, levels of p47 and p67 subunits of NADPH oxidase were found to be significantly elevated in blood PMNs following ATP treatment (Figure 4C,D). Similar induction of p67 was detected after ATP treatment (100 μM) and this induction was markedly suppressed when blood PMNs were pretreated with BAPTA-AM (2 or 5 μM) or Gö6983 (2 or 5 μM) for 20 min (Figure 4E), indicating that upregulation of these subunits may be involved in the ATP-mediated induction of ROS. Importantly, pretreatment with Trolox (an ROS scavenger, 10 or 20 μM) or apocynin (an NADPH oxidase inhibitor, 10 or 20 μM) significantly suppressed the induction of PAD4 and CitH3 and reduced the amounts of extracellular DNA-NE complex in blood PMNs culture media (Figure 4F–H), confirming the importance of ROS in ATP-mediated induction of NETosis. Similar induction in ROS production and its suppression by A438079, BAPTA-AM, or Gö6983 were observed also in BzATP-treated blood PMNs (Appendix A). These results demonstrate that ATP-P2X7R signaling enhances ROS production in neutrophil and suggest that this process plays a critical role in NETosis induction.

### 2.5. ATP Released Following NMDA-Induced Neuronal Death Induces NETosis

Next, we investigated if extracellular ATP accumulated following MCAO is derived from neurons following NMDA-induced neuronal death and whether this ATP is capable of inducing NETosis (Figure 5A). Treatment of primary cortical cultures with NMDA (100 μM) resulted in ATP accumulation in NMDA-conditioned medium (NCM), with ATP levels gradually increasing until 120 min post-treatment (Figure 5B). Since HMGB1, another DAMP molecule, which was previously shown to induce NETosis [7] was detected in NCM 120 after min of NMDA treatment (Figure 5C), we used NCM collected at 90 min to investigate the function of ATP in NCM. Induction of PAD4 in blood PMNs began after 2 h of NCM treatment (Figure 5D) and subsequent increase of CitH3 levels was also detected after 4 h and peaked at 6 h (Figure 5E). However, pretreatment with A438079 (20 μM), Go6983 (2 μM), or BAPTA-AM (2.5 μM) for 20 min before NCM treatment significantly suppressed the inductions of PAD4, CitH3, and dsDNA levels in NCM-treated blood PMNs (Figure 5F–H), indicating that P2X7R, Ca^2+^, and PKCα play critical roles. Additionally, preincubation of cells with apocynin (20 μM) or Trolox (20 μM) suppressed NCM-mediated inductions of PAD4, CitH3, and dsDNA release (Figure 5F–H). Importantly, ATP was observed to accumulate in culture media of blood PMNs after treating NCM for 3 h, with the amount further increased after 9 h (Figure 5I), indicating that ATP is a component of NETs extruded as a result of NETosis. Taken together, these results indicate that ATP is responsible for the induction of NETosis by excitotoxicity-induced neuronal death and that it is released from blood PMNs as a component of NETs.

### 2.6. ATP-P2X7R-Mediated Induction of NETosis in the Ischemic Brain

We next examined whether ATP-P2X7R and down-stream signaling molecules we have identified mediate ATP-mediated induction of NETosis in vivo. When we administered apyrase, which catalyzes the hydrolysis of ATP to AMP and inorganic phosphate, at 6 h post-MCAO (0.2 U/g, i.v.) (Figure 6A), inductions of PAD4 and CitH3 observed 12 h post-MCAO were significantly suppressed in blood PMN and the inductions observed in cortical penumbra 24 h post-MCAO were also significantly suppressed (Figure 6B–D). Immunofluorescent staining with anti-CitH3 antibody showed that the numbers of CitH3^+^ cells increased in cortical penumbra at 24 h post-MCAO, however, they were significantly decreased in apyrase-administered MCAO animals (Figure 6E,F). Together these results demonstrate a crucial role of ATP in NETosis in the ischemic brain. In addition, administration of apocynin (2.5 mg/kg) or Trolox (2.5 mg/kg) 9 h post-MCAO resulted in a marked suppression of the inductions of PAD4 and CitH3 observed at 12 h post-MCAO (Figure 6G). Moreover, the administration of Trolox or apocynin (2.5 mg/kg, i.v.) 3 h after BzATP administration (Figure 6A) suppressed BzATP-mediated enhancements of CitH3 and PAD4 in cortical penumbra at 12 h post-MCAO (Figure 6h), demonstrating the critical role of ROS in ATP-P2X7R-mediated NETosis in the ischemic brain. Immunofluorescence staining of blood PMNs isolated 12 h post-MCAO with anti-CitH3 antibody further confirmed NETosis induction following MCAO and an important role of ROS in this process (Figure 6I,J). Taken together, these results indicate that ATP accumulated after cerebral ischemia induces NETosis in the ischemic brains and within blood vessels.

## 3. Discussion

This study demonstrates that ATP accumulates after cerebral ischemia and plays a critical role in the induction of NETosis. ATP-mediated NETosis was confirmed to occur in blood PMNs and BM-PMNs through a mechanism that involves P2X7R-mediated Ca^2+^ influx, PKCα activation, and increased ROS production, leading to the subsequent upregulation and activation of PAD4 (Figure 7). Although ATP has been shown to amplify NET formation induced by calcium ionophore or PMA [20], this is the first report showing that ATP alone can induce NETosis.

It has been reported that extracellular ATP is hydrolyzed by membrane-bound ecto-ATPases, which convert ATP to ADP and AMP [28]. It has been reported that during the cerebral ischemia, the conversion of extracellular ATP to ADP and AMP to adenosine proceeds immediately following extracellular release and that these conversions may regulate extracellular ATP levels [29]. In vitro, we observed an increased accumulation of ATP in primary cortical neuron culture media following NMDA treatment (Figure 5B). However, it should be noted that in the ischemic brain, additional cell types such as microglia and astrocytes may contribute to extracellular ATP levels, and extracellular ATP might be hydrolyzed by ectonucleotidases. Even with these factors taken into consideration, we demonstrated that ATP accumulation in NCM was high enough to stimulate P2X7R on neutrophils and induce NETosis in a P2X7R-, Ca^2+^-, and ROS-dependent manner. Moreover, a marked suppression of CitH3 induction in both blood PMNs and cortical penumbra following MCAO by administrating apyrase strongly support a crucial role of ATP in the NETosis induction in the ischemic brain.

It is important to note that, as is the case for HMGB1 [7], ATP is also a component of the cellular content extruded during NETosis. Extruded ATP from NETosed neutrophils may exacerbate the inflammatory response in the ischemic brain by further recruiting and activating neighbor neutrophils and other immune cells. Accordingly, it would appear that a vicious cycle mediated by ATP is established that aggravates the inflammatory response following permanent MCAO. Interestingly, it was recently reported that P2Y6 receptor, whose main ligand are UDP and UTP, is involved in the aggravation of NETosis in gout associated with monosodium urate (MSU) [30]. Since large amounts of ectonucleotides (such as ATP, ADP, UDP, UTP, and adenosine) were reported to be released during NETosis, a vicious cycle may exist between NETosis induction and autocrine response to other ectonucleotides via purinergic receptors such as P2X7 and P2Y6. Furthermore, considering that macrophages also contribute to the aggravation of NETosis by releasing cytokines, as was reported in MSU-induced gout [31], this cycle may expand in a paracrine manner.

We report here that ATP-mediated induction of NETosis was similarly stimulated by a prototypic P2X7R agonist BzATP and was prevented by A804598 (a selective P2X7R antagonist), demonstrating the importance of P2X7R in these processes. In evaluating the underlying mechanism, we found PAD upregulation to play a critical role in P2X7R-mediated induction of NETosis, since PAD4 and CitH3 levels were further enhanced by BzATP administration in MCAO animals. Detrimental effects of P2X7R were demonstrated in animal models of cerebral ischemia [32] and intracerebral hemorrhage [33,34,35], with proposed involvement of blood brain barrier disruption [33], NLRP3 inflammasome-mediated aggravation of inflammation [34], and the induction of secondary brain injury [35] as underlying mechanisms. The current study adds a novel putative mechanism; NETosis induction. However, we cannot exclude the possibility that ATP indirectly induces NETosis via P2X7R-mediated NLRP3 inflammasome activation, causing cell death and NET release [16,34] or via pannexon channel activation [36]. Further studies are needed.

ATP-P2X7R-mediated PAD4 induction shown in the present study raised another possible novel function of ATP-P2X7R, which is PAD4-mediated post-translational protein modification. PAD4 initiates chromatin decondensation during NETosis by catalyzing the conversion of arginine to citrulline, especially for histone H3 or H4 [37]. Since PAD4 is a Ca^2+^-dependent enzyme that requires elevated Ca^2+^ levels for activation [38], P2X7R-induced intracellular Ca^2+^ flux also contribute to enhance PAD4 activity. Therefore, both expression and activity of PAD4 may be enhanced in the ischemic brain by extracellular release of ATP caused by acute and massive necrotic cell death. PAD4 was reported to play a critical role in several pathologies, especially in autoimmune diseases such as rheumatoid arthritis, which is genetically linked to PAD4 [39]. PAD4 also produces citrullinated neoantigens to promote autoimmune diseases [40]. Recently, CitH3 was reported to induce microvascular leakage and endothelial barrier dysfunction [41] and excessive NET formation was shown to promote vascular leakage [42] and induce epithelial and endothelial cell death due to the cytotoxic effects mediated by histones localized in NETs [43]. Further studies are, therefore, warranted to determine the possibility that ATP-initiated PAD4 induction/activation causes damage to endothelial or other cells, and consequently disrupt the BBB in the ischemic brain.

With regard to the specific subtype of PKC activated during NETosis, PKCβ was shown to be activated in PMA-induced-NETosis [44]. However, we found that PKCβ was not activated in blood PMNs by BzATP (Appendix A). Evaluation of ROS indicates that ATP-mediated NETosis is dependent on NADPH oxidase, acting in downstream of P2X7R, Ca^2+^, and PKCα. In blood PMNs, we observed that both ATP and BzATP induce the p47 and p67 subunits of NADPH oxidase, which is in accordance with previous reports of P2X7R-mediated upregulation of these subunits in obese mice with inflammatory liver injury [45]. However, since numerous molecular mechanisms for underlying NADPH oxidase modulation by P2X7R have been reported, including the translocation of p47 or p67 subunits to the plasma membrane [45,46] and activation of NLRP3 inflammasome [44], regulation of ROS presents a multifaceted target for the reduction of ATP-induced NETosis in sterile disease.

When cells are damaged, ectonucleotides such as ATP, UTP, and UDP are released and accumulate in extracellular space [47], and are subsequently recognized as danger signals by immune cells, evoking various responses via purinergic receptors [48]. In addition to ATP and HMGB1, uric acid is another DAMP molecule, which was reported to induce NETosis. However, NETosis induced by uric acid or MSU differed from ATP-induced NETosis by being NADPH oxidase-independent [49]. Additionally, MSU, a salt generated readily from ionized uric acid, in synovial fluid causes gout-induced NET formation by human neutrophils through P2Y6 receptor [30]. Further studies are clearly needed to explore the similarities and differences between NETosis induced by different DAMP molecules. In summary, our results indicate that ATP accumulates in the ischemic brain, induces NETosis, and might mediate a cycle between NETosis and neuronal damage that might aggravate inflammation and potentiate subsequent brain damage.

## 4. Materials and Methods

### 4.1. Reagents

A438079 (P2X7 antagonist), Go6983 (PKC inhibitor), or BAPTA (intracellular Ca^2+^ chelator) was purchased from Tocris Bioscience (Bristol, UK). ATP, BzATP, Phorbol 12-myristate 13-acetate (PMA, PKC activator), apocynin (NADPH oxidase inhibitor), Trolox (ROS scavenger), or apyrase were purchased from Sigma Aldrich (St. Louis, MO, USA). ATP (5~50 mg/kg), BzATP (5 mg/kg), Apyrase (0.2 U/kg) were administered intravenously in 0.3 mL PBS, and apocynin (2.5 mg/kg) or Trolox (2.5 mg/kg) was dissolved in DMSO and administered intravenously in 0.3 mL PBS after 9 h of pMCAO.

### 4.2. Surgical Procedure for Permanent MCAO

All animals used in this study were treated with consideration to minimizing pain. The study protocol was reviewed and approved by the INHA University-Institutional Animal Care and Use Committee (INHA-IACUC; Approval Number INHA-141124-337-2, Approval date: 24 November 2014) prior to the commencement of the study. Additionally, the study was conducted in accord with the Guide for the Care and Use of Laboratory Animals published by the National Institute of Health (2010) and its presentation complies with ARRIVE (Animal Research: Reporting In Vivo Experiments) guidelines [50]. Male Sprague-Dawley (SD) rats (8 weeks old, 230–250 g body weight) were purchased from Orient Bio Inc. (Gyeonggi, South Korea). Animals were kept in a humidity and temperature-controlled room under a 12 h dark/light cycle, with free access to food and tap water for a week prior to the start of the experiments. Permanent focal cerebral ischemia was generated as previously described [7]. Briefly, rats (9 weeks old) were anesthetized with isoflurane (3% isoflurane induction, 2% maintenance) in a 30/70% oxygen/nitrous oxide mixture. The right common carotid artery (CCA), internal carotid artery (ICA), and external carotid artery (ECA) were exposed through a neck midline incision. A monofilament nylon suture (4-0; AILEE, Busan, Korea) was slowly inserted into the ECA and advanced ~20 mm from the carotid bifurcation, until resistance was felt. For sham surgeries, CCA, ICA, and ECA were exposed without an insertion of a suture. A laser Doppler flowmeter (Periflux System 5000; Perimed, Jarfalla, Sweden) was used to monitor regional cerebral blood flow. During surgery, animals were placed on a heating pad kept at 37.0 ± 0.5 °C. Animals were randomly allocated to 10 treatment groups, as follows; (1) the MCAO group; PBS-treated MCAO animals (*n* = 31), (2) the MCAO + BzATP group; BzATP-treated MCAO animals (*n* = 6); (3) the ATP group; ATP-treated animals (*n* = 4); (4) the BzATP group; BzATP-treated animals (*n* = 3); (5) the MCAO + apyrase group; apyrase-treated MCAO animals (*n* = 8); (6) the MCAO + apocyanin; apocyanin-treated MCAO animals (*n* = 9); (7) the MCAO + trolox; trolox-treated MCAO animals (*n* = 9); (8) the MCAO + BzATP + apocyanin; BzATP and apocyanin-treated MCAO animals (*n* = 2); (9) the MCAO + BzATP + trolox; BzATP + trolox-treated MCAO animals (*n* = 2); (10) the Sham group; animals underwent surgery but were not subjected to MCAO (*n* = 21). 

### 4.3. Isolation of Circulating Neutrophils

Circulating neutrophils were isolated from the collected blood samples by gradient density centrifugation using Histopaque solution (Sigma Aldrich, St. Louis, MO, USA), as previously described [9]. Briefly, Histopaque 1077 (3 mL) was layered on Histopaque 1119 (3 mL) in a 15 mL polypropylene tube and collected blood (3 mL) was carefully layered on top of the mixture (Histopaque 1077/1019). The layered system comprising the three components was then centrifuged at 400× *g* for 30 min at room temperature (RT). The first ring of cells (mononuclear cells) was discarded. The second ring (neutrophils) was transferred to a fresh 15 mL polypropylene tube containing PBS-BG (phosphate buffered solution, 0.1% bovine serum albumin, and 10% glucose) and centrifuged at 600× *g* for 10 min at RT. To remove residual red blood cells, the resulting pellet was resuspended in 3 mL of PBS-BG, decanted onto Histopaque-1119 (3 mL), and centrifuged at 600× *g* for 15 min at RT. The neutrophil ring was then transferred to a fresh 15 mL polypropylene tube containing PBS-BG and centrifuged at 600× *g* for 10 min at RT. The resulting pellet was resuspended in RPMI (Gibco BRL, Gaithersburg, MD, USA) containing 1% FBS.

### 4.4. Isolation of Bone Marrow Neutrophils

Bone marrow (BM) was flushed from femurs and tibiae of hind legs using a syringe fitted with a 22 G needle. BM cells were collected in a 15 mL polypropylene tube containing PBS-BG and centrifuged at 600× *g* for 10 min at RT. Pellets were resuspended in 45% Percoll and gently layered on top of a Percoll gradient (50%, 55%, 62%, and 81%). After centrifugation at 600× *g* for 30 min at RT, the cell band that formed between the 81% and 62% Percoll layers was collected in a 15 mL polypropylene tube containing PBS-BG. To remove residual red blood cells, cells from the obtained band were layered onto 3 mL of Histopaque 1119 and centrifuged at 600× *g* for 20 min at RT. Collected cells (1 × 10^7^ cells) were resuspended in α-MEM (Gibco BRL, Gaithersburg, MD, USA). All experiments were performed after attachment of cells (~2 h) on the slides.

### 4.5. Measurement of Blood ATP Levels

Blood ATP levels were measured using a luminescent ATP detection assay kit (Abcam, Cambridge, UK). Blood samples were quickly transferred to microcentrifuge tubes and centrifuged at 15,000× *g* for 1 min at 4 °C. The upper aqueous phase was then diluted with 20 volumes of distilled water, and 100 μL of this diluted extract was added to the luciferin/luciferase mixture provided in the kit. Luminescence was measured using a luminometer (Perkin Elmer, Waltham, MA, USA). Blood ATP levels were read off a calibration curve.

### 4.6. Immunofluorescence Staining

Isolated blood PMNs were cytocentrifuged (ThermoFisher Scientific, Waltham, MA, USA) at 500 rpm for 8 min at RT and then fixed with 4% paraformaldehyde (PFA, Sigma Aldrich, St. Louis, MO, USA). Cytospin slides were then blocked with 1% normal goat serum for 30 min and incubated overnight with anti-CitH3 antibody (ab18956-100; Abcam, Cambridge, UK) at 4 °C. DNA was stained with DAPI (4′,6-diamidino-2-phenylindole; Sigma Aldrich, St. Louis, MO, USA) to confirm the nuclear localization of CitH3 and nuclei. Slides were observed under a fluorescence microscope (Axioplan 2; Zeiss, Oberkochen, Germany). To prepare brain tissue, animals were sacrificed 24 h after surgery and brains were isolated and fixed with 4% paraformaldehyde (PFA; Sigma Aldrich, St. Louis, MO, USA) by transcardiac perfusion and then stored in the same solution overnight at 4 °C. Brain sections (40 μm) were prepared using a vibratome. Primary antibodies for anti-CitH3 (ab18956-100; Abcam, Cambridge, UK) and anti-RECA (MCA970; Bio-Rad, Hercules, CA, USA) were diluted 1:200. Brain sections were counterstained with DAPI (4′,6-diamidino-2-phenylindole; Sigma Aldrich, St. Louis, MO, USA) to visualize nuclei, and observed under a fluorescence microscope (Axioplan 2; Zeiss, Oberkochen, Germany). The numbers of CitH3-positive cells in 0.16 mm^2^ (0.4 × 0.4 mm) were scored.

### 4.7. Immunoblotting

Blood and BM-PMNs were washed twice with cold PBS and lysed in RIPA buffer containing 50 mM Tris-HCl (pH 7.4), 0.5% Triton X-100, 0.5% NP-40, 0.25% sodium-deoxycholate, 150 mM NaCl, and Complete Mini Protease Inhibitor Cocktail tablets (1 tablet per 10 mL) (Roche, Basel, Switzerland). Cell lysates were centrifuged at 25,000× *g* for 15 min at 4 °C and supernatants were loaded onto 6–12% SDS PAGE gels. Anti-CitH3 (ab18956), anti-PKCα (ab32376), anti-PKCβII (ab38279), and anti p67 (ab67282) primary antibodies were purchased from Abcam (Cambridge, UK), while anti-phospho PKCα (SC377565) and anti-p47 (SC17845) were purchased from Santa Cruz (Santa Cruz, CA, USA). Anti-phospho PKCβII (7371) and GAPDH (2118) primary antibodies were purchased from Cell Signaling Technology (Danvers, MA, USA) and anti-PAD4 (GTX54621) was purchased from GeneTex (Irvine, CA, USA). Primary antibodies were diluted 1:2000-10,000. Immunoblotting signal was detected using a chemiluminescence kit (Merck Millipore, Darmstadt, Germany).

### 4.8. Preparation of Mixed Neuron-Glia Cultures

Mixed cortical cells were prepared from embryonic day 15.5 (E15.5) mouse cortical tissue, as previously described [7]. Briefly, cortical brain tissue was dissected and dissociated in MEM media using Pasteur pipettes drawn to a smaller inner diameter under a flame. Dissociated cortical cells were plated at a density of 4 × 10^5^ cells per well on a poly-d-lysine (100 μg/mL) and laminin (100 μg/mL)-coated 24 well plate. Cells were maintained in MEM containing 5% FBS, 5% horse serum, 21 mM glucose, and 2 mM glutamine for 7 days. Glial cell proliferation was inhibited by addition of cytosine arabinoside (10 μM) to MEM containing 10% horse serum and 21 mM glucose on days in vitro (DIV) 7 and the medium was changed every other day thereafter. Cultures were used for experiments between DIV 12 and 14.

### 4.9. Treatment of Neutrophils with NMDA-Conditioned Media (NCM)

Mixed cortical cells (4 × 10^5^/well) were treated with NMDA (100 μM) (Sigma, St. Louis, MO, USA) in MEM for 30 min, washed with MEM, and incubated for 90 min in MEM. NMDA-conditioned medium was collected, and neutrophils (2 × 10^6^) were treated with NCM (a mix of NCM from four wells) for 6 h. Neutrophils were pretreated with A438079 (20 μM), Go6983 (2 μM), BATPA (2.5 μM), apocynin (20 μM), or Trolox (20 μM) for 20 min prior to treatment with NCM.

### 4.10. Quantification of Cell-Free DNA

Levels of cell-free DNA in culture medium were measured using the Quant-iT PicoGreen double-stranded DNA (dsDNA) assay kit (Invitrogen, Carlsbad, CA, USA). Supernatants from cultured neutrophils were collected, mixed with assay kit reagent, and fluorescence was measured using a spectrofluorometer (Molecular Devices, Sunnyvale, CA, USA) at excitation/emission of 480/540 nm. DNA concentrations were calculated using a standard curve prepared using kit-supplied DNA. Data were normalized relative to NET-DNA (ng/mL) released by stimulating cells with ATP, BzATP, or PMA.

### 4.11. Quantification of NETs-DNA Complexes in Cell Supernatant

To measure neutrophil elastase (NE)-DNA complexes, anti-NE (1:1000, ab21595, Abcam, Cambridge, UK) was coated onto 96-well microtiter plates (100 μL per well) overnight at 4°C. The plate was washed two times with PBS, and then blocked with 1% BSA (100 μL per well) for 60 min at room temperature. The plate was again washed three times with PBS, and the supernatant (100 μL) obtained from ATP-treated cell culture was added to the plate and incubated for overnight at 4°C. After washing three times with PBS, Sytox green (100 μL, 1:15,000) was added to each well. NE-DNA complex were quantified using the Quant-iT PicoGreen double-stranded DNA (dsDNA) assay kit (Invitrogen, Carlsbad, CA, USA) according to the manufacturer’s instructions and fluorescence was measured in Gen5 microplate reader (Bio Tek, Winooski, VT, USA) at emission/excitation wavelength of 504/523 nm.

### 4.12. Intracellular Calcium Levels

Intracellular Ca^2+^ levels were measured in blood neutrophils using Fluo-4-AM. Briefly, 2 × 10^6^ cells/mL were incubated in HBSS-Mg^2+^ (calcium free, 1% FBS) media containing 4 μM Fluo-4-AM for 15 min in a CO_2_ incubator. After washing with HBSS (Ca^2+^-free), neutrophils were resuspended in RPMI (1% FBS) and seeded in 24-well plates. Fluo-4-AM fluorescence was measured using a JuLi^TM^ Stage (NanoEntek, Seoul, Korea) and fluorescence intensities were quantified using ImageJ (http://rsbweb.nih.gov/ij/) (accessed on 16 October 2020).

### 4.13. Quantification of Reactive Oxygen Species

Blood PMNs or BM-PMNs were incubated in αMEM (20 mM) containing 1 μM of CM-H2DCFDA (a mixture of 5-(and-6)-chloromethyl-2′,7′-dichlorodihydrofluorescein diacetate; Invitrogen, Carlsbad, CA) for 30 min. Cells were washed twice with PBS, and fluorescence images were visualized under JuLi^TM^ Stage (NanoEntek, Seoul, Korea). Quantitative analysis of the immunofluorescence data was carried out using ImageJ (http://rsbweb.nih.gov/ij/) (accessed on 16 October 2020).

### 4.14. Statistical Analysis

Sample sizes for animal experiments were determined using G power calculation 3. 1. 9. 7 (Germany) (http://www.gpower.hhu.de/ with the levels of significance of 5% and a minimum power of 80%. Analysis of variance (ANOVA) followed by the Newman–Keuls test was used to determine the significance of differences. Results are presented as mean ± SEM, with difference with *p*-values < 0.05 considered statistically significant.

## Figures and Tables

**Figure 1 ijms-21-07668-f001:**
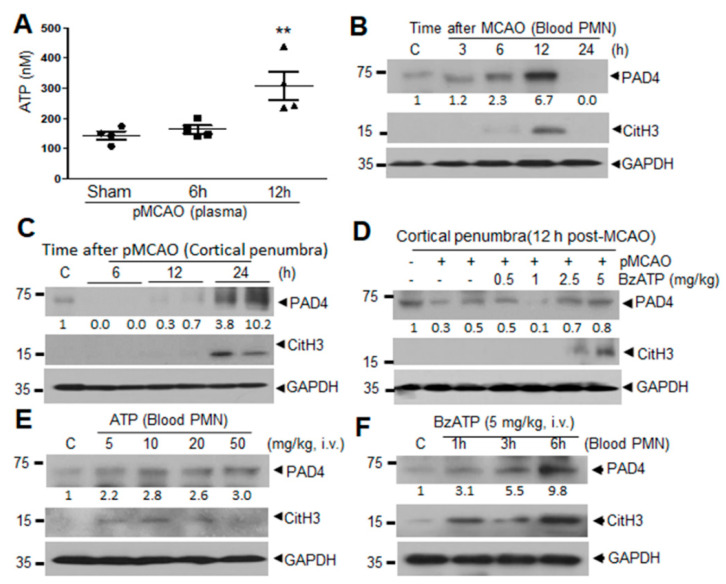
Adenosine triphosphate (ATP) accumulation and ATP-mediated induction of PAD4 and CitH3 in blood neutrophils and cerebral cortical tissue after MCAO in male Sprague-Dawley (SD) rats. (**A**) ATP levels were measured using ATP luminescence kit in plasma collected at 6 and 12 h post-MCAO. (**B**,**C**) Levels of PAD4 and CitH3 in blood neutrophils (blood PMNs) or in cortical penumbra were examined at 3, 6, 12, and 24 h post-MCAO by immunoblotting. (**D**) BzATP (0.5, 1, 2.5, or 5 mg/kg, i.v.) was administered 6 h post-MCAO and PAD4 and CitH3 levels were examined in cortical penumbra at 12 h post-MCAO by immunoblotting. (**E**) ATP (5, 10, 20, or 50 mg/kg, i.v.) was administered to normal rats and PAD4 and CitH3 levels were examined in blood PMNs isolated 4 h after ATP administration by immunoblotting. (**F**) BzATP (5 mg/kg, i.v.) was administered to normal rats and PAD4 and CitH3 levels were examined in blood parenchyma and in circulating neutrophils (PMNs) isolated 1, 3, or 6 h after BzATP administration by immunoblotting. Immunoblots are representative of 2 or 3 independent experiments. Numbers under the blot indicate band intensities. ATP levels were presented as mean ± SEM (*n* = 4). ** *p* < 0.01 versus sham controls.

**Figure 2 ijms-21-07668-f002:**
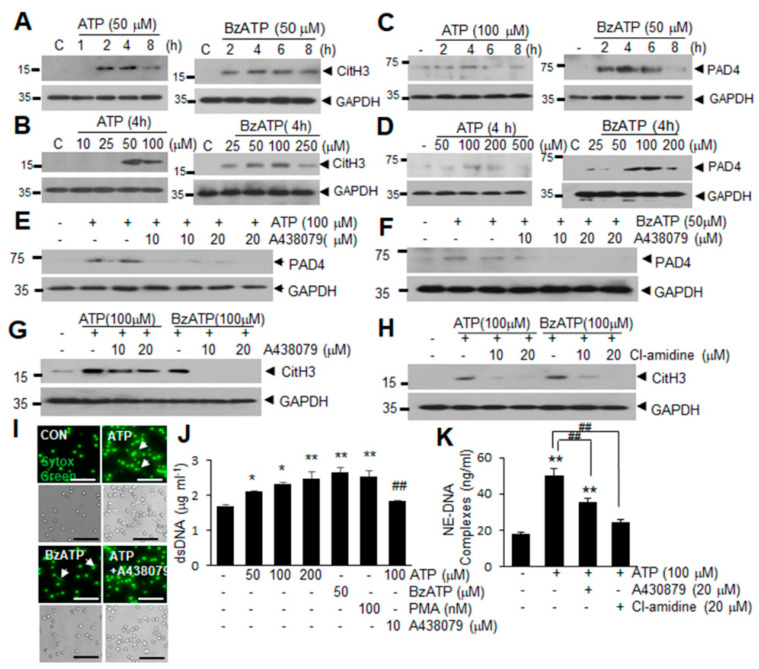
ATP induced PAD4 and CitH3 upregulation in neutrophils and promoted NETosis in a P2X7R-dependent manner. (**A**–**D**) Blood PMNs were treated with 50 μM of ATP or BzATP for the indicated duration (**A**,**C**) or a range of doses of ATP or BzATP for 4 h (**B**,**D**). Levels of CitH3 (**A**,**B**) and PAD4 (**C**,**D**) were determined by immunoblotting. (**E**,**F**) Blood PMNs were pretreated with A438079 (10 or 20 μM) for 20 min before treatment with 100 μM ATP (**E**) or 50 μM BzATP (**F**) for 4 h. PAD4 levels were subsequently assessed by immunoblotting. (**G**,**H**) Blood PMNs were pretreated with A438079 (10 or 20 μM) (**G**) or Cl-amidine (10 or 20 μM) (**H**) for 20 min prior to treatment with 100 μM ATP or BzATP for 4 h. CitH3 levels were determined by immunoblotting. Immunoblots are representative of 2~4 independent experiments. (**I**–**K**) Blood PMNs were pretreated with A438079 (10 μM) or Cl-amidine (20 μM) for 20 min prior to treatment with ATP (50, 100, or 200 μM) or with BzATP (50 μM) for 4 h. dsDNA release was visualized by staining with Sytox Green (**I**) and the amounts of free dsDNA (**J**) or DNA complexed with neutrophil elastase (NE-DNA) (**K**) were assessed using Quant-iT PicoGreen dsDNA reagent. Arrows in I indicate the released DNA and scale bars in represent 50 μm. Results are presented as mean ± SEM (*n* = 3). * *p* < 0.05, ** *p* < 0.01 versus the PBS-treated control, ## *p* < 0.01 versus ATP (100 μM) only-treated cells.

**Figure 3 ijms-21-07668-f003:**
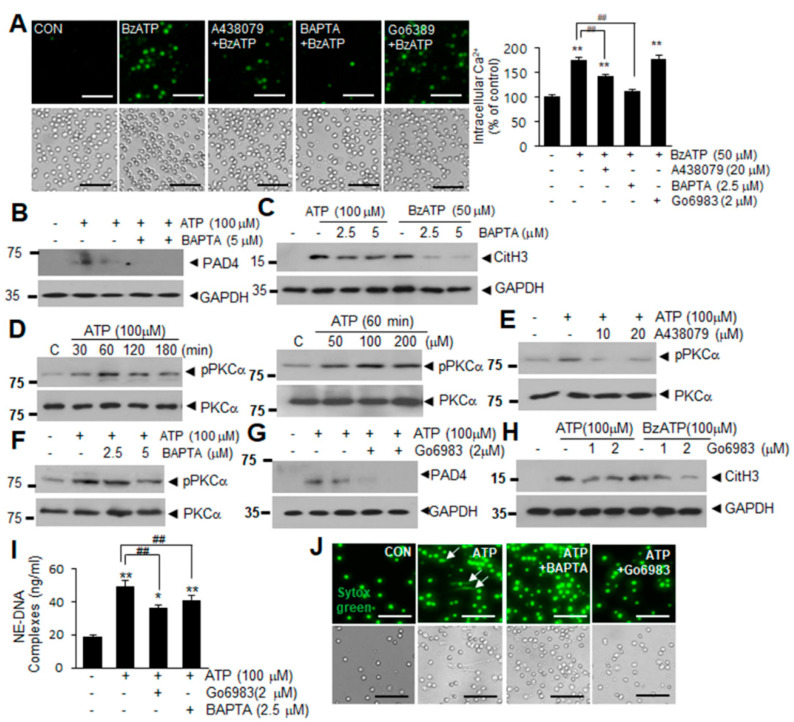
Intracellular Ca^2+^ influx and PKCα activation in ATP-P2X7R-induced NETosis. (**A**) Blood PMNs were pretreated with A438079 (10 μM), BAPTA (2.5 μM), or Go6983 (1 μM) for 20 min prior to treatment with BzATP (50 μM) for 30 min. Intracellular Ca^2+^ levels were visualized by Fura 2-AM and quantified using ImageJ (http://rsbweb.nih.gov/ij/, accessed on 16 October 2020). Quantification of intracellular Ca^2+^ was normalized relative to PBS-treated controls. (**B**,**C**) Blood PMNs were pretreated with BAPTA (2.5 or 5.0 μM) for 20 min prior to treatment with ATP (100 μM) or BzATP (50 μM) for 4 h. Levels of PAD4 (**B**) or CitH3 (**C**) were subsequently determined by immunoblotting. (**D**) Blood PMNs were treated with ATP (100 μM) for the indicated duration or with 50, 100, or 200 μM of ATP for 30 and levels of PKCα and phospho-PKCα were examined by immunoblotting. (**E**,**F**) Blood PMNs were pretreated with A438079 (10 or 20 μM) or with BAPTA (2.5 or 5 μM) for 20 min prior to treatment with ATP (100 μM) for 30 min, then the levels of PKCα and phospho-PKCα were subsequently determined by immunoblotting. (**G**,**H**) Blood PMNs were pretreated with Go6983 (1 or 2 μM) for 20 min prior to treatment with ATP (100 μM) or BzATP (100 μM) for 4 h, then PAD4 or CitH3 levels were subsequently determined by immunoblotting. Immunoblots are representative of 2~4 independent experiments. (**I**,**J**) Blood PMNs were pretreated with BAPTA (2.5 μM) or Go6983 (2 μM) for 20 min prior to treatment with 100 μM of ATP for 12 h, with amounts of extracellular DNA complexed with neutrophil elastase (NE-DNA) subsequently assessed using Quant-iT PicoGreen dsDNA assay kit (**I**) and visualized using Sytox Green (**J**). Arrows in j indicate the released DNA and scale bars represent 50 μm. Results are presented as mean ± SEM (*n* = 3). ** *p* < 0.01, * *p* < 0.05 versus the PBS-treated control, ## *p* < 0.01 between indicated group.

**Figure 4 ijms-21-07668-f004:**
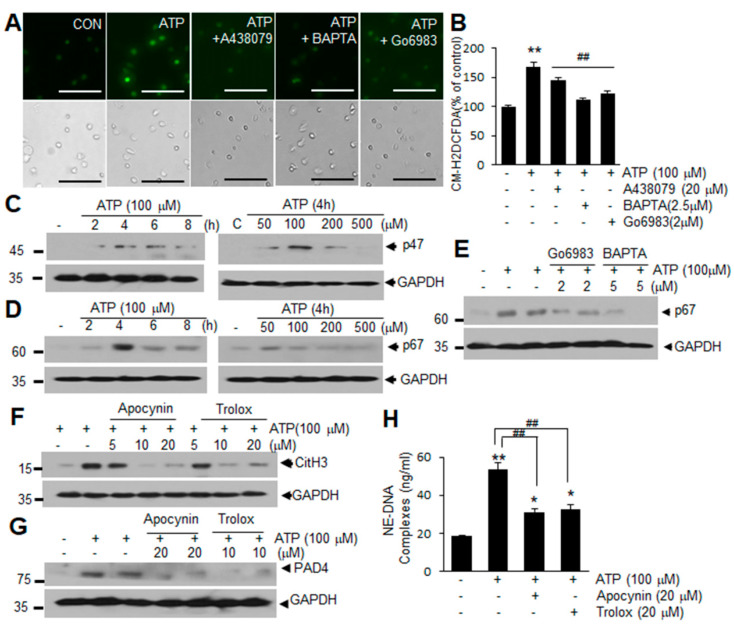
Upregulation of ROS production via P2X7R in ATP-induced NETosis in blood PMNs. (**A**,**B**) Blood PMNs were pretreated with A438079 (10 μM), BAPATA (2.5 μM), or Go6983 (1 μM) for 20 min prior to treatment with ATP (100 μM) for 2 h. Intracellular ROS generation was visualized using CM-H2DCFDA and analyzed using ImageJ (http://rsbweb.nih.gov/ij/, accessed on 16 October 2020). (**C**,**D**) Levels of p47 or p67 in blood PMNs were determined after 2, 4, 6, or 8 h of treatment with ATP (100 μM), or after 4 h of treatment with a range of concentrations of ATP. (**E**) Blood PMNs were pretreated with BAPATA (2 or 5 μM) or Go6983 (2 or 5 μM) for 20 min prior to treatment with ATP (100 μM) for 4 h, with p67 level were subsequently assessed by immunoblotting. (**F**,**G**) Blood PMNs were pretreated with 5, 10, or 20 μM of apocynin or Trolox for 20 min prior to treatment with ATP (100 μM) for 4 h, with CitH3 (**F**) and PAD4 (**G**) levels were subsequently assessed by immunoblotting. Immunoblots are representative of 2~4 independent experiments. (**H**) Blood PMNs were pretreated with 20 μM of apocynin or Trolox for 20 min prior to treatment with ATP (100 μM) for 12 h, with the amounts of extracellular DNA complexed with neutrophil elastase (NE-DNA) subsequently assessed using Quant-iT PicoGreen dsDNA assay kit. Scale bars in A represent 50 μm. Results are presented as mean ± SEM (*n* = 3). ** *p* < 0.01, * *p* < 0.05 versus the PBS-treated controls, ## *p* < 0.01 versus ATP only-treated cells.

**Figure 5 ijms-21-07668-f005:**
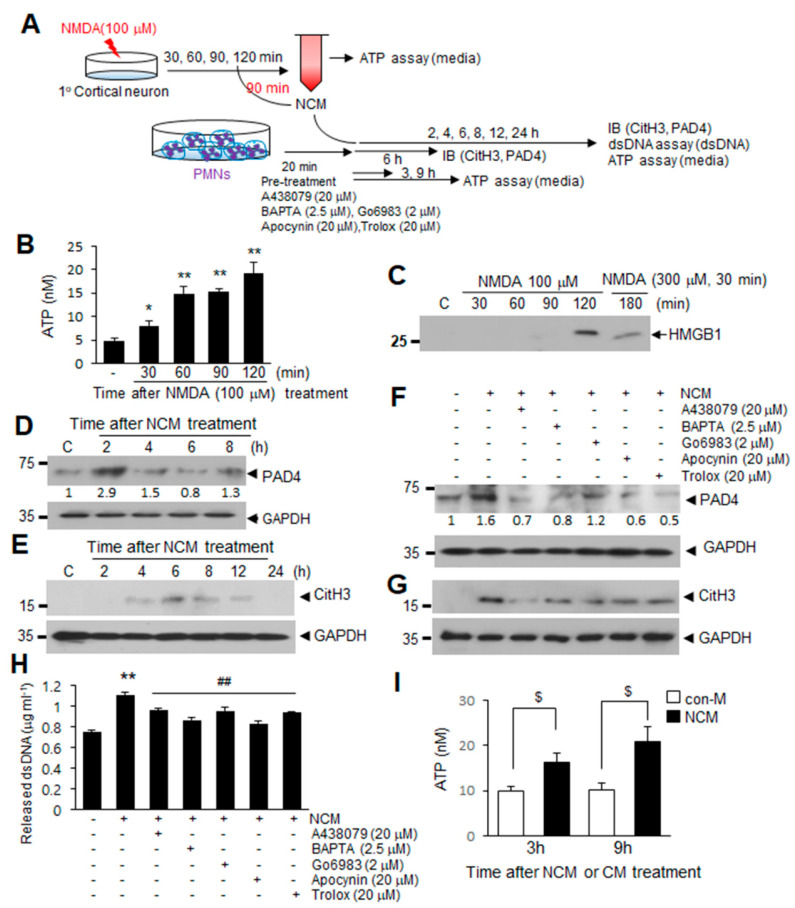
ATP release after excitotoxic neuronal cell death induces NETosis via a P2X7R-Ca^2+^-PKC signaling pathways. (**A**) Primary cortical neuron cultures were treated with 100 μM NMDA for various duration. NMDA-conditioned media (NCM) was prepared after 90 min of NMDA treatment and used to treat blood PMNs. (**B**) Primary cortical neuron cultures were treated with 100 μM NMDA for 30, 60, 90, or 120 min, and the ATP levels in primary cortical neuron culture media were subsequently measured using an ATP assay kit. (**C**) HMGB1 levels in culture media were determined after treating primary cortical neurons with NMDA (100 μM) for 30, 60, 90, or 120 min by immunoblotting. (**D**,**E**) Levels of PAD4 and CitH3 in NCM-treated blood PMNs were determined at the indicated times by immunoblotting. (**F**,**G**) Blood PMNs were pretreated with A438079 (20 μM), BAPTA (2.5 μM), Go6983 (2 μM), apocynin (20 μM), or Trolox (20 μM) for 20 min prior to treatment with NCM for 6 h, with the levels of PAD4 and CitH3 subsequently determined using immunoblotting. Immunoblots are representative of 2~4 independent experiments. (**H**) dsDNA release was measured after pretreating blood PMNs with A438079 (20 μM), BAPTA (2.5 μM), Go6983 (2 μM), apocynin (20 μM), or Trolox (20 μM) for 20 min prior to treatment with NCM. (**I**) Blood PMNs were treated with NCM or NMDA-untreated control media (con-M) for 3 or 9 h, and ATP levels in culture media of blood PMNs were determined using an ATP assay kit. Results are presented as mean ± SEM (*n* = 3). ** *p* < 0.01, * *p* < 0.05 versus the PBS-treated controls, ## *p* < 0.01 versus NCM only-treated cells, ^$^
*p* < 0.05 between indicated group. Numbers under the blot indicate band intensities.

**Figure 6 ijms-21-07668-f006:**
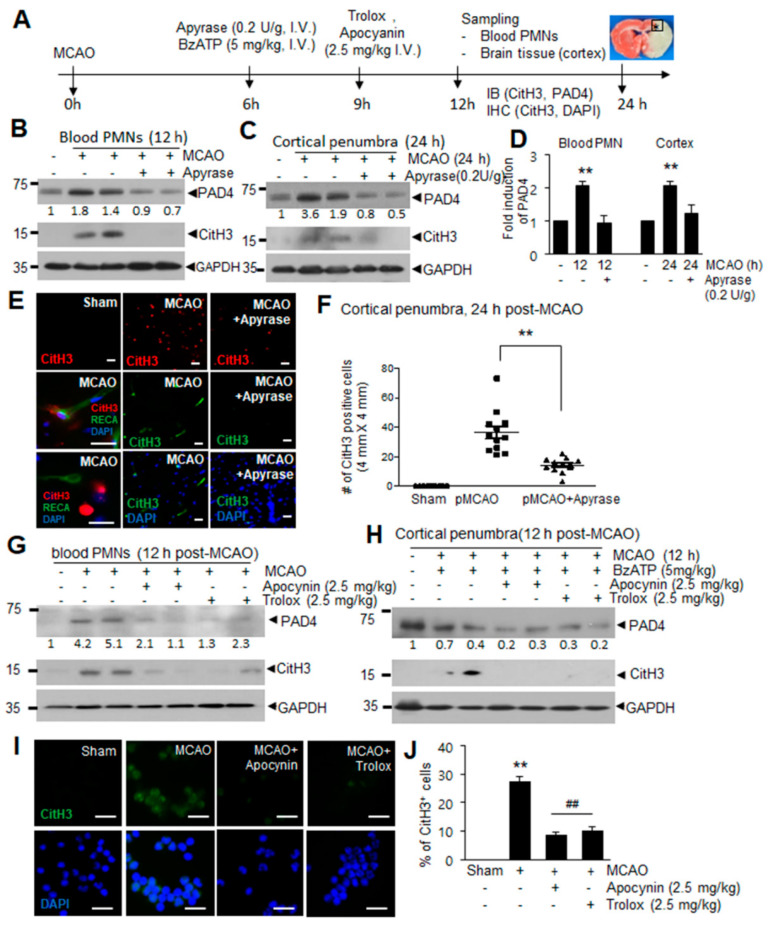
ATP-P2X7R mediates NETosis induction in ischemic brains. (**A**) Apyrase (0.2 U/g, i.v.) and BzATP (5 mg/kg, i.v.) was administered at 6 h post-MCAO and Trolox (2.5 mg/kg, i.v.) and apocynin (2.5 mg/kg, i.v.) was administered at 9 h post-MCAO. Immunoblotting and immunofluorescence staining was carried out at 12 or 24 h post-MCAO. (**B**–**F**) PAD4 and CitH3 levels in blood PMNs or cortical penumbras of ischemic hemispheres were determined at 12 or 24 h post-MCAO, respectively, by immunoblotting with or without apyrase (0.2 U/g, i.v.) treatment at 6 h post-MCAO (**B**–**D**), and immunofluorescence staining of brain tissues sections with CitH3 antibody was carried out at 24 h post-MCAO (**E**,**F**). (**G**,**I**,**J**) Apocynin (2.5 mg/kg) or Trolox (2.5 mg/kg) was administered 9 h post-MCAO and levels of PAD4, CitH3, and NET formation in blood PMNs purified at 12 h post-MCAO were assessed by immunoblotting (**G**) or immunofluorescence staining with CitH3 antibody (**I**,**J**), respectively. (**H**) PAD4 and CitH3 levels in cortical penumbra of the ischemic hemisphere were determined at 12 post-MCAO after treating BzATP at 6 h post-MCAO and with or without Trolox (2.5 mg/kg, i.v.) and apocyanin (2.5 mg/kg, i.v.) administration at 9 h post-MCAO. Immunoblots are representative of 2 or 3 independent experiments. Scale bars in (**E**,**I**) represent 50 μm. Results are presented as mean ± SEM (*n* = 3). ** *p* < 0.01 versus sham control, ## *p* < 0.01 versus MCAO. Numbers under the blot indicate band intensities.

**Figure 7 ijms-21-07668-f007:**
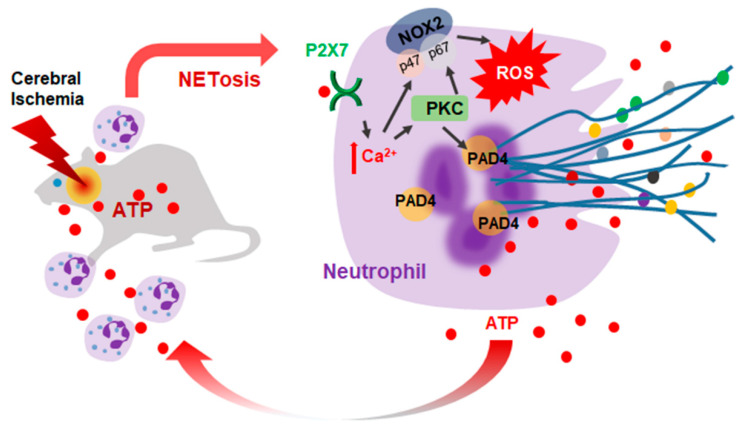
Diagram of the ATP-mediated NETosis induction in cerebral ischemia. ATP is released from neurons and glia during the acute phase of cerebral ischemia. ATP induces NETosis in the ischemic brain wherein P2X7R, Ca^2+^, PKC, and ROS plays important roles and PAD4 initiates chromatin decondensation. Extruded NET components induce brain damage and ATP included in expelled NETs accelerates this process. NOX, NADPH oxidase; PAD4, protein deiminase 4; ROS, reactive oxygen species.

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
