# Peer review of "Adenosine Triphosphate Accumulated Following Cerebral Ischemia Induces Neutrophil Extracellular Trap Formation"

_ijms, 2020, doi:10.3390/ijms21207668_

Round 1

Reviewer 1 Report

The authors suitably addressed all the comments, and performed additional experiments with ATP to induce NETosis in blood PMNs. The revised manuscript is now acceptable for publication in IJMS.

Author Response

Thank you 

Reviewer 2 Report

The authors should address the following points:

  1. By measuring NE-DNA-complexes, authors provide stronger evidence that NETs are produced following ATP treatment, and that these the release of structures can be inhibited by the PAD4 inhibitor, Cl-amidine (Fig. 2K). With respect to these data, several minor issues should be addressed.
    1. The method for this technique and the Quant-iT Pico-Green assay need to be untangled in the methods.
    2. Can the authors speculate as to why the concentration of dsDNA and NE-DNA are 1000-fold different? Is this expected, or does it suggest that only a small fraction of the extracellular dsDNA are ‘bona fide’ NETs?

  1. The number of true biological replicates (N = ?) are still unclear. This reviewer would define replicates as independent experiments which have been conducted with the same variables. It appears that this study also includes technical replicates (e.g., where 10 nM is used twice) and experiments with different variables (times, concentrations, and cell types) to account for the number of experimental replicates. The authors should be explicit as to how many biological replicates were used.

  1. Several specific comments in the original review have not been addressed:
    1. (R2, point 2) Why were higher concentrations of ATP were used to treat blood neutrophils than what were found in the blood following MCAO?
    2. (R2, point 4) How much supernatant (or micrograms of lysate) was loaded into gels for immunoblots?

  1. There are a few missing or blurry labels
    1. The symbol above some of the bars is not completely visible (Fig. 5H, for example).
    2. A438079 is missing from Fig. 5A
    3. In Fig. 7 NOX should be NOX2.

Author Response

  1. By measuring NE-DNA-complexes, authors provide stronger evidence that NETs are produced following ATP treatment, and that these the release of structures can be inhibited by the PAD4 inhibitor, Cl-amidine (Fig. 2K). With respect to these data, several minor issues should be addressed.
    • The method for this technique and the Quant-iT Pico-Green assay need to be untangled in the methods.

Response: We rewrote the Method section in the revised manuscript.

  • Can the authors speculate as to why the concentration of dsDNA and NE-DNA are 1000-fold different? Is this expected, or does it suggest that only a small fraction of the extracellular dsDNA are ‘bona fide’ NETs?

Response: We authors appreciate the reviewer’ question. Thanks to the question, we found our critical mistake. When we measured DNA concentration, we took 100 ml of supernatant and obtained total amount of DNA. Because the amount of DNA was represented as “ng/ml” in the bar graphs, we had to multiply the value by 10 but we didn’t. We corrected them in the revised manuscript. However, the concentration of dsDNA and NE-DNA are still 50~-fold different. Although we can speculate that not all of the extruded DNA are present to be bound to neutrophil elastase, we cannot tell the exact ratio.   

  1. The number of true biological replicates (N = ?) are still unclear. This reviewer would define replicates as independent experiments which have been conducted with the same variables. It appears that this study also includes technical replicates (e.g., where 10 nM is used twice) and experiments with different variables (times, concentrations, and cell types) to account for the number of experimental replicates. The authors should be explicit as to how many biological replicates were used.

Response: We understand reviewer’s point. In our manuscript, number of replicates varies and dose- or temporal test experiments, it was two. For example in Figure 2, n=2 (A-D, G, H), n=4 (E,F, two sets containing duplicate), and n=3 (J,K). The number of replicates for immunoblots are between 2 and 4. We mentioned this in the Figure legend of each figure in the revised manuscript.

  1. Several specific comments in the original review have not been addressed:

3-1.(R2, point 2) Why were higher concentrations of ATP were used to treat blood neutrophils than what were found in the blood following MCAO?

Response: In our initial dose-testing experiment, we found that at least 50 mM of ATP are needed to induce PAD4 or CitH3. I am afraid that we cannot explain the underlying reason, it might be due to the differences between in vivo and in vitro conditions.

3-2.(R2, point 4) How much supernatant (or micrograms of lysate) was loaded into gels for immunoblots?

Response: We loaded 10 mg of total protein for each well.

  1. There are a few missing or blurry labels

4-1.The symbol above some of the bars is not completely visible (Fig. 5H, for example).

Response: We corrected them in the revised manuscript.

  • A438079 is missing from Fig. 5A

Response: We corrected it in the revised manuscript.

  • In Fig. 7 NOX should be NOX2.

Response: We corrected it in the revised manuscript.

This manuscript is a resubmission of an earlier submission. The following is a list of the peer review reports and author responses from that submission.

Round 1

Reviewer 1 Report

This study showed that ATP released from the ischemic brain induced NETs formation in the blood PMN and in the cortical penumbra. It also showed P2X7R-Calcium ion-ROS dependent NETs formation induced by ATP using a P2X7R antagonist, a calcium chelater, a pan-PKC inhibitor and antioxidants, step by step. This study was well designed to deduce the conclusion from the results. There are some comments to be addressed and described by the authors, to understand the significance of the results of this study.

  1. In the ischemic brain, ATP and markers for NETs formation were measured in the cortical penumbra. The reason why cortical penumbra was used in this study has not been described. Was the ischemic injury including infarction reduced in cortical penumbra by drugs blocking NETs formation?
  2. The number of animals used in each experiment per treatment or group is required
  3. The rationale how the doses of drugs and administration time were determined in in vivo studies should be described (ex, BzATP, Agyrase, Trolox. Apocyamin).
  4. All immunoblots of western blotting were representative. How many sets of experiment were performed independently?
  5. In Fig.3, the first figure showed the results from BzATP treated blood PMNs, but the other results in B to J were derived from ATP treated cells. Also in Fig. 4, A and B showed the results of ATP treated cells, and the others in C-H were BzATP treatment. It was confusing to understand following the results. Was there any reason to test ATP and BzATP interchangeably in the experiments?
  6. In Fig.5I, if cytotoxicity of NCM was determined, the conclusion of vicious cycle of ATP release from NETosis would be strengthened.
  7. In Fig.6H, PAD4 and CitH3 were measured in cortical penumbra 12 h after MCAO in rats treated with BzATP to boost NETs formation. Considering the result shown in Fig.1C, it seems that the experiment of 24 h after MCAO in rats without BzATP treatment would be more physiologically relevant. Please describe the reason to use rats treated with BzATP instead of the same of 6G experimental paradigm with examination at 24 h after MCAO. Also, in Fig. 1D and 6H, PAD4 densities in negative control animals were much higher than those in the ischemic brain, which were different compared with that of control animals shown in Fig.1C and 6C. Please confirm which result was correct.

Reviewer 2 Report

Following up on their observation that plasma ATP levels increased after middle cerebral artery occlusion (MCAO) in rats, authors of this manuscript demonstrate a link between extracellular ATP levels, increased levels of PAD4, and citrullinated histones in tissue from the cortical penumbra and in circulating neutrophils. Authors clearly show that ATP acting on the P2XR7 receptor causes an increase in intracellular calcium and phosphorylation of PKCa, which were all required for inducible PAD4 expression and citrullinated histones.  Although this pathway is very interesting, several concerns merit the authors attention:

  1. There are several major flaws with respect to measuring NET production, and thus, conclusions drawn about NETs are questionable. Since extracellular DNA can be indicative of general cell death, NETs are often defined using several markers, including MPO, histones, and a dependence on PAD4 and elastase (Reviewed in Yousefi et al., 2019, European Journal of Immunology). Although DNA release was measured using Sytox green and PicoGreen, no additional markers of NETs were included in their analysis. There is also no evidence that the DNA release required PAD4 or histone citrullination because correlation does not equal causation. In other work histone citrullination was not required for NET production (Tsourouktsoglou et al., Cell Reports, 2020). Authors did show that Cl-amidine inhibited histone citrullination following ATP-treatment (Fig 2H), but did not look at DNA release in parallel. Lastly, the staining pattern of sytox green-stained control cells is unexpected. Sytox green should be impermeable to living cells, and in Figure 2I, 3J, and supplemental Fig. 1I, the dye stains control cells brightly. Authors should confirm both neutrophil purity and viability following isolation.

  1. Authors state that following MCAO ATP levels increase until 12 hours and then decline. Authors should show these data and discuss why they selected higher concentrations of ATP and longer incubation periods to stimulate neutrophils. For some incubation periods (>12 h), neutrophils may undergo apo-necrosis. Were control neutrophils matched for each time point?

  1. If NETs are in fact generated, then the question that remains is it really ATP directly causing NET production? Alternatively, could ATP act through the NLRP3 inflammasome to cause cell death and NET release? Assessing caspase 1 activity or gasdermin cleavage would provide evidence for a direct or indirect pathway. Or is ATP is causing cell death via activation of purinergic receptors, recruitment of pore-forming connexins, and leakage of DAMPs? Authors should also discuss the discrepancy between their work and the study from Sofoluwe et al. (2019 Science Reports) that demonstrates that ATP has no direct effect on NET production, but rather amplifies production of PMA- or A23187-induced NETs.

  1. With respect to the immunoblots, authors should: (1) clarify the number of experimental replicates, (2) quantify band intensities, (3) include blots for total histone, minimally following ATP treatment, and (4) define how much supernatant was loaded. Since many of the conclusions were based on the immunoblot data, quantification and statistical analysis would be especially convincing.

  1. It is unclear from the introduction what unanswered question the study will address or what hypothesis is being tested. Authors should consider condensing the review of prior work, focusing on mechanisms PAD4-dependent NET formation and ATP signaling. Authors should also be careful to imply that NETs cause inflammation, and that NETs are part of a “vicious” cycle.Many examples exist where NETs are not pro-inflammatory, and no data was provided to suggest that NETs caused additional damage following MCAO.

Minor:

  1. The title should be changed to reflect the largely in vitro approach used in this study.
  2. GAPDH appears to resolve either above or below 35 kDa. Please confirm the correct positioning of these labels.
  3. In Fig. 5B, is the control PBS, or spent media? Spent media from cortical neutrons would be a better control for this experiment.
  4. Statistical analysis in Fig. 5H is questionable (n = 12), but is in fact 3 mice with 4 images per mouse. The three mice represent the values that should be used in the ANOVA.
  5. Fig 6D. shows fold induction of what?
  6. There are a couple typos – (1) Fig.6G is missing Trolox +; (2) line 489 missing c; DNase 1 vs. I (line 60 and 64).